# MRI-Compatible Microcirculation System Using Ultrasonic Pumps for Microvascular Imaging on 3T MRI

**DOI:** 10.3390/s22166191

**Published:** 2022-08-18

**Authors:** Ju-Yeon Jung, Dong-Kyu Seo, Yeong-Bae Lee, Chang-Ki Kang

**Affiliations:** 1Department of Health Science, Gachon University Graduate School, Gachon University, Incheon 21936, Korea; 2Department of Radiological Science, College of Health Science, Gachon University, Incheon 21936, Korea; 3Department of Neurology, Gil Medical Center, Gachon University College of Medicine, Incheon 21565, Korea

**Keywords:** microvascular imaging, MRI-compatible flow system, micro-tube phantom, piezoelectric pump

## Abstract

The diagnosis of small vessel disease is attracting interest; however, it remains difficult to visualize the microvasculature using 3 Tesla (T) magnetic resonance imaging (MRI). Therefore, this study aimed to visualize the microvascular structure and measure a slow flow on 3T MRI. We developed a microcirculation system using piezoelectric pumps connected to small tubes (0.4, 0.5, 0.8, and 1.0 mm) and evaluated various MR sequences and imaging parameters to identify the most appropriate acquisition parameters. We found that the system could image small structures with a diameter of 0.5 mm or more when using a 1 m-long tube (maximal signal intensity of 241 in 1 mm, 199 in 0.8 mm, and 133 in 0.5 mm). We also found that the highest signal-to-noise ratio (SNR) appeared on 2-dimensional time-of-flight low-resolution imaging and that the flow velocity (10.03 cm/s) was similar to the actual velocity (11.01 cm/s in a flowmeter) when velocity encoding of 30 cm/s was used in a 0.8 mm-diameter tube. In conclusion, this study demonstrates that a microcirculation system can be used to image small vessels. Therefore, our results could serve as a basis for research on vessels’ anatomical structure and pathophysiological function in small vessel disease.

## 1. Introduction

Various brain disorders, such as stroke and brain aging, are caused by cerebral stenosis or occlusion with abnormal blood circulation in the brain, resulting in abnormal vascular function and hemodynamics [1]. Therefore, detection technologies, such as high-resolution brain imaging, are being developed for the early diagnosis or prevention of brain diseases. Despite advances in imaging technology, microvascular detection technology is underdeveloped.

A lacunar stroke occurs in territories supplied by the perforating arteries. It can also be an indirect cause of dementia, cerebral stroke, and other severe neuropsychiatric disorders [2]. Therefore, detecting microvascular diseases is important for the early diagnosis and prevention of cerebrovascular diseases. However, diagnosing lacunar infarction using the brain imaging equipment is difficult for several reasons. First, lacunar infarction occurs in the microvessels, such as the lenticulostriate arteries, which have 0.1–1.28 mm diameters [3]. Second, it is difficult to observe small vessels using the conventional imaging acquisition parameters of 3 Tesla (T) magnetic resonance imaging (MRI) because of insufficient sensitivity and imaging resolution. While common infarction or aneurysm that occurs in cerebral vessels, such as the circle of Willis, enables an early diagnosis before a lesion appears, lacunar infarction is usually observed in territories of small vessels (<15 mm) [4].

Recent studies identifying the pathophysiology of lacunar infarction used ischemic regional images of lacunar infarction because of the limitations of small vessel imaging in 3T MRI [5]. For the early diagnosis and treatment of lacunar infarction and small vessel disease (SVD), small vessel imaging studies, including vascular structural imaging and/or microcirculation feature identification, are necessary. Therefore, the conventional blood circulation system and acquisition parameters on 3T MRI should be properly modified to enable small vessel imaging.

Studies on magnetic resonance angiography (MRA) used blood circulation systems and manufactured vascular phantoms [6,7,8,9,10]. However, these vascular structural and flow studies were mainly performed using relatively large arterial vascular models [6,11,12,13] (Table 1). Furthermore, the circulation pump used in the previous flow studies featured tubes that were >5 m long because the pump must be located outside the magnetic field due to the magnetic force [6]. For this reason, some small vessel phantom studies confirmed small vascular structures without a flow delivery system [14]. Therefore, previous conventional flow phantom systems are not suitable for visualizing small vessels (Table 1). Furthermore, such studies could not generate a continuous flow within the small vessel phantom, making the evaluation of the characteristics of normal microvascular blood flow challenging.

Microvessel research has mainly been conducted using ultra-high-field 7T MRI [15,16]; however, 7T is still difficult to commercialize for diagnosis because of its nonuniform radiofrequency (RF). Therefore, small vessel imaging should be conducted using 3T MRI. In this study, we developed a small vessel flow system using piezoelectric pumps that can supply adequate and continuous blood flow to small vessels, even within a magnetic field. Using the developed microcirculation system, we could visualize the small vessel structure, measure its slow flow using 3T MRI, and suggest the proper imaging sequence and parameters for small vessel imaging.

## 2. Materials and Methods

### 2.1. Flow System Configuration

The flow system consisted of a pump, a controller, a microvascular phantom, a flowmeter, a water reservoir, and two types of tubes, 1 m and 6 m long (Figure 1), that were used to test the signal changes according to the microtube length. The piezoelectric pump was located inside the MRI system, and the controller and flowmeter were placed outside the MRI system in the control room.

The pump used a diaphragmatic ultrasonic piezoelectric motor with a diameter of 21 mm (SDMP-320, TAKASAGO, Tokyo, Japan); the pump size was 33 × 33 × 5.5 mm^3^, and the inner and outer diameters of the inlet and outlet for the tubing were 1.8 mm and 2.8 mm, respectively. An ultrasonic pump uses a piezoelectric phenomenon in which motion occurs owing to the polarization of piezoelectric ceramics when a driving voltage is applied and is suitable for use in MRI because it does not generate a magnetic field by non-magnetic materials, such as piezoelectric ceramics [17,18,19]. Therefore, the ultrasonic piezoelectric pump inside the MRI was operated at a certain distance (approximately 80 cm) from the bottom of the radiofrequency head coil, which was located on the magnet during scanning.

The controller operating at 5 V direct current (SDMP-320, TAKASAGO, Tokyo, Japan) was a transformer delivering the driving voltage to the ultrasonic piezoelectric pumps, and the output voltage and frequency were adjustable between 60 and 300 V and 1.0 and 60.0 Hz, respectively. In this study, the output voltage of the piezoelectric pump was 250 V, and the output frequency was 40 Hz. The controller was connected to the pumps using a 5 m-long wire to operate them.

Four silicon tubes (0.4, 0.5, 0.8, and 1.0 mm) were prepared for microvascular phantoms, and two tubes (0.4 and 0.5 mm or 0.8 and 1.0 mm) were simultaneously placed on a transparent acrylic stand measuring 210 × 210 × 60 mm^3^ in a parabolic pattern. The distance between the two tubes was 2.5 cm, and the flow was fed through two piezo-ultrasonic pumps. The inlet was a single tube with an inner diameter of 1.5 mm and length of 30 cm. The water reservoir was connected to the inlet and outlet tubes, allowing inflow and outflow.

To measure the flow rate, a series-connected thermal mass flowmeter (SLS-1500, Sensirion AG, Staefa, Switzerland) using the small heat exchange difference between the liquid and the tube during flow through the heated tube was used. The measurable range of flow rate was 0–40 mL/min, the measurable minimum flow rate unit was 5 μL, and the driving voltage ranged from 4 to 6 V. A universal serial bus (USB) port was connected to measure the values. In this study, the flow rate was acquired at a sampling interval of 4 ms and recorded using the software (USB RS485 Sensor Viewer, Sensirion AG, Staefa, Switzerland) provided by the manufacturer.

### 2.2. MRA Experimental Protocols

The experiment was performed using a clinical 3T MRI scanner (Siemens Vida, Erlangen, Germany) with a 20-channel head coil, and MRA images were obtained using two-dimensional (2D) low- and high-resolution (Low and High, respectively) time-of-flight (TOF) MRA, 3D High TOF MRA, and 2D phase-contrast (PC) MRA sequences. The scan parameters, including the scan times of all the sequences, are listed in Table 2.

### 2.3. Flow Analysis in the Microvascular Phantom

The flow velocity of the microvascular phantom was measured using a flowmeter, and the Reynolds number was calculated for the flow types of all tubes. The fluid used was purified water; thus, a kinematic viscosity of 10^−6^ m^2^/s was used [20]. The Reynolds numbers were 23.32, 28.85, 93.84, and 91.2 at 0.4, 0.5, 0.8, and 1.0 mm and were in the range of the laminar flow.

### 2.4. Comparison of Images by Tube Size and Length and MR Sequences

The performance of the flow system, depending on the size and length of the tubes, was tested using coronal maximum intensity projection (MIP) images from 3D High TOF MRA. The diameters of the tubes were 0.4, 0.5, 0.8, and 1.0 mm, and their lengths were 1 m and 6 m. For the same imaging parameters of the 3D High TOF MRA, differences were compared in the coronal maximal intensity projection (MIP) images along the diameter and length of the tube.

In addition, to verify the appropriate MR parameter for the microflow system, we compared the signal-to-noise ratio (SNR) of three different TOF sequences. In this data analysis, signal intensity profiles were examined on axial images of 2D Low, 2D High, and 3D High TOF MRA. In addition, the SNR depending on the diameter of the 1 m tube was also calculated for each sequence, with 15 consecutive regions of interest (ROIs) being selected. The average value of the two largest SNRs was further analyzed depending on the imaging sequence.

### 2.5. Reproducibility of Images Obtained in the Flow System

The reproducibility of the flow supply through the flow system was determined by analyzing the coefficients of variation (CoV) of seven axial images obtained repetitively with the same imaging parameters in 2D PC MRA. The CoV aids in evaluating the accuracy of the data and reproducibility of the equipment without the influence of the measured units [21,22,23]. To evaluate the reproducibility of the seven repetitive measurements, the MIP image for each dataset was reconstructed to compare the signal intensities. In the MIP images of 0.8 mm and 1.0 mm tubes in the 1 m tube, the ROI was selected to cover 30 voxels, and the average signal intensity and standard deviation were obtained within the ROI. The CoV was calculated using the following formula:$$\mathrm{Coefficient}\;\mathrm{of}\;\mathrm{Variation}\;\left(\mathrm{CoV}\right)=\frac{\mathrm{Standard}\;\mathrm{Deviation}}{\mathrm{Average}\;\mathrm{Intensity}}\times 100\%$$

### 2.6. Comparison of the Actual Velocity Measured by the Flowmeter with the Velocity Measured in the PC MRA Image

The actual flow velocity was measured using a flowmeter. The velocity calculated from the axial image of 2D PC MRA was compared with that measured using a flowmeter. The recording flow rate of the flowmeter was 25 times per second. The flow velocity calculation was performed on the axial MIP image of 2D PC MRA using the following formula:Velocity (cms)=Signal Intensity of ROI × VENC4096
where VENC is the velocity encoding, including 10, 20, and 30 cm/s (named VENC 10, VENC 20, and VENC 30, respectively). All data analyses were performed using ImageJ analysis software (National Institutes of Health, Bethesda, MD, USA).

## 3. Results

### 3.1. Comparison of Images by Tube Size and Length

In 3D High TOF coronal images, the signals of the microvascular phantoms were compared according to the tube diameter and length (Figure 2). The signal intensity decreased as the diameter of the tube decreased and it also decreased from the 1 m tube to the 6 m tube (Table 3). The signals of both 1 m and 6 m tubes with 0.4 mm diameter were not clearly visualized in these images (Figure 2).

The SNRs of each sequence are compared in Figure 3. The signal intensity and SNR decreased in all the sequences as the tube diameter decreased. The image obtained in the 2D Low TOF sequence showed the highest SNR for all diameters (1.0 mm = 3.93; 0.8 mm = 3.03; 0.5 mm = 1.35; 0.4 mm = 0.76) compared to those of the other sequences; further information for all sequences is shown in Table 4. The image obtained in the 3D High TOF sequence showed the lowest SNR for all diameters (1.0 mm = 0.46; 0.8 mm = 0.29; 0.5 mm = 0.10; 0.4 mm = 0.09).

### 3.2. Reproducibility of the Blood Flow Supply from the Piezoelectric Pump

Seven images were repeatedly acquired on the same phantom to examine the reproducibility of the flow system, especially in the 0.8 mm and 1.0 mm tubes (Figure 4). Their average signal intensities and their CoVs were 619.3 ± 32.69 and 624 ± 25.7, and 5.27% and 4.13% in the 0.8 mm and 1.0 mm tubes, respectively (Table 5).

### 3.3. Evaluation of the Optimal Imaging Parameters and Tube Size in the Developed System

The CoV of actual velocities was measured to examine the variation in the velocities within the MRI environment according to the VENC. For all tube diameters, CoV exhibited a low variation rate. Among them, the lowest CoV appeared in the 0.4 mm tube, and the highest appeared in the 1.0 mm tube (e.g., 0%, 1.94%, 2.65%, and 3.48% for 0.4, 0.5, 0.8, and 1.0 mm tubes, respectively).

For all tube diameters, the difference between the actual flow velocity and the 2D PC MRA velocity was smallest in VENC 30. Furthermore, VENC 30 of the 0.8 mm tube was the most similar to that measured in the flowmeter. A difference of only 0.98 cm/s was observed in the 0.8 mm tube with VENC 30 (10.03 and 11.01 cm/s, respectively) (Figure 5 and Table 6). The highest difference was observed in the 0.4 mm tube with VENC 10 (1.13 and 5.70 cm/s, respectively).

## 4. Discussion

This study aimed to visualize the microvascular structure and measure a slow flow using 3T MRI by developing a microcirculation system that works properly with 3T MRI. For this purpose, we used MR-compatible ultrasonic piezoelectric pumps to generate a continuous and stable flow delivery system.

We confirmed the microphantom structure and obtained a reasonable flow signal in the microtubes. Previously, the assessment of microvascular structures presented various problems in imaging the microphantom in 3T MRI, because conventional electric pumps work limitedly in the MRI; therefore, the length of the tubes was long, allowing a decreased flow rate because of high flow pressure. Previous vascular studies used a 9 mm-diameter flow phantom with a 5 m-tube flow system (Table 1) [6]. Therefore, the internal pressure increased with the use of a long tube, which decreased the flow rate and further reduced the signal intensity, making it difficult to image small vascular structures. In this study, the signal differences for different tube lengths were clearly visible. We imaged 0.4–1.0 mm-diameter micro phantoms with 1 m and 6 m tubes (Table 3 and Figure 2). According to our results, most microdiameter tubes, except the 0.4 mm tube, had a higher signal intensity with 1 m length than with 6 m length. The signal intensity increased up to 26% from the 6 m to the 1 m tube (signal intensity = 190 and 241, respectively) with 1.0 mm diameter and up to 33% from the 6 m to the 1 m tube (signal intensity = 100 and 133, respectively) with 0.5 mm diameter. In particular, the structural image of the 0.5 mm tube could be visualized in the 1 m tube but not in the 6 m tube, suggesting that the minimal diameter of the microtube phantom that could be imaged was approximately 0.5 mm. We also differentiated the inflow effect between the 1 m and the 6 m tube images (Figure 2). The inflow effect is characterized by a higher signal closer to the entrance of the imaging region and a lower signal farther from the entrance. This is because the protons excited with the RF pulse from a previous slice do so again with the next RF pulse before the signal is recovered; this is called the saturation effect. That is, the signal intensity gradually decreases as the distance from the entrance of the imaging region increases and it is further degraded across longer tube lengths. Therefore, it can be concluded that shorter tubes are more effective in a microcirculation system.

Therefore, using MRI-compatible pumps is essential because the tube length should be as short as possible to obtain a high signal intensity.

In addition to the diameter and length of the tubes, microvascular flow images are also affected by the pulse sequences (Figure 3 and Table 4). We measured the SNRs to compare the image quality of the three TOF sequences. We found that the 3D TOF High images showed the lowest SNR values for all types of tubes, and the 2D TOF Low images showed the highest SNR values. Furthermore, the 2D TOF Low images had a better SNR owing to the small tube size. It is well known that 2D TOF MRA is sensitive to a slow flow compared with 3D TOF MRA and that 3D TOF MRA images have a relatively high resolution and small pixel sizes, causing a low SNR in the vessels [24,25,26].

We were able to confirm the stability of continuous signal intensity for the 0.8 and 1.0 mm tubes, as shown in Figure 4 and Table 5. We scanned the 2D PC MRA seven times in succession to examine the continuous and consistent flow signals. The results showed that the CoVs of the axial images obtained with seven repetitive datasets were 5.27% and 4.13% for the 0.8 and 1.0 mm tubes, respectively, proving that the ultrasonic piezoelectric pumps within the MRI system could supply a stable flow (Figure 4 and Table 5). The CoV suggests that the lower the value within 30%, the better the precision of the method or equipment [22,23].

Figure 5 and Table 6 show the actual flow velocity measured using a flowmeter compared with that calculated from the 2D PC MRA signals. The actual flow velocities measured by the flowmeter were constant at VENC 10, VENC 20, and VENC 30, and the PC MRI flow velocity also had a low variation rate for all tube diameters. However, the actual flow velocity decreased from the 0.8 to the 1.0 mm tube. This indicates that the power of the piezoelectric pumps used was not sufficient for a 1.0 mm tube.

In 2D PC MRA, the velocity can be affected by VENC value and tube diameter. If the VENC is too high compared to the actual velocity, the slow flow signal in a small tube appears as a noise signal. Therefore, selecting a VENC value suitable for the actual flow velocity is important to prevent this phenomenon. Therefore, here we evaluated VENC 10, 20, and 30, which were also previously used for microvessels, such as the lenticulostriate arteries [27,28], and were similar to those measured with a flowmeter. As a result, the VENC 30 image of the 0.8 mm-diameter tube showed the lowest difference from the actual flow velocity because the pumps with current specifications worked better with tubes smaller than 0.8 mm- than with 1.0 mm-diameter tube, vascular aliasing was prevented, and the vascular signal intensity was much higher than the noise [29,30,31]. The MRA velocities of microvascular flow were similar to those reported in previous studies [27,28]. According to Schnerr (2017), the mean velocity of the lenticulostriate arteries is 8.2 cm/s, very similar to that we measured with our 1.0 mm tube (8.21 cm/s). Another previous study reported a mean velocity of 9.6 cm/s [28], which is also accordant with our results (10.03 cm/s in an 0.8 mm tube). These results support the feasibility of the developed microcirculation system and imaging parameters evaluated here for small vessel imaging. However, the 0.4 mm diameter tube showed a high difference from the actual flow velocity. This is because the velocity of the 0.4 mm tube seemed too low for all VENCs, owing to the small proton content and a small contribution to the signal within a voxel. This problem also appeared in the structural image of the 0.4 mm tube (Table 3). Therefore, the 0.8 mm tube model was the most suitable for our microphantom flow system with a piezoelectric pump.

This study has some limitations. First, the VENC values are important to determine the flow velocity through MRA images. Further research is required to determine the optimal values. Furthermore, we adjusted only the VENC value among various imaging parameters; thus, additional experiments are still necessary to find optimal sequences and imaging parameters for the accurate measurement of microvascular velocity, such as to further adjust the flip angle, repetition time, and echo time [32]. Second, it was difficult to confirm the image of the 0.4 mm-diameter tube using our microflow system. Therefore, it is necessary to develop piezoelectric pumps with better performance for the study of microvessels, such as pumps with higher power or a combination of pumps. Furthermore, microvascular imaging should be further evaluated in a more realistic small vessel environment, including blood or blood-like fluids, small vessel disease models (e.g., stenosis and aneurysm), and/or branches. Further studies could elucidate the pathophysiological mechanisms in actual small vessel disease, such as lacunar infarction, by developing microvascular models such as lenticulostriate arteries.

In this study, ultrasonic piezoelectric pumps for the flow system were located inside the MRI system and continuously supplied a stable flow to the microvascular phantoms that had adjustable tube sizes and lengths. Further usage of the proposed microcirculation system would be in the lacunar infarction model and is expected to play an important role in devising methods for diagnosing and treating microvascular diseases.

## 5. Conclusions

This study demonstrated that microvascular images can be obtained using 3T MRI with an MRI-compatible microcirculation system. The system using ultrasonic piezoelectric pumps could be placed inside a 3T MRI and provided a stable flow with a CoV ranging from 4% to 5%. This study showed that images of microvascular tubes of 0.5 mm or more could provide sufficient signal intensity using 3T MRI. Furthermore, this study demonstrated that a 1 m length of the tube improved the signal intensity in the microcirculation system. The flow rate of the 0.8 mm microvascular phantom calculated in the VENC 30 image was similar to the flow rate obtained with a flowmeter.

This result suggests that the length of the tube must be as short as possible. Thus, a 0.8 or 0.5 mm-diameter tube is recommended for microvascular measurements in the developed flow system. Therefore, it is essential to use a pump compatible with MRI for this purpose. Furthermore, 2D or low-resolution sequences and VENC 30 are recommended for measuring a slow flow in the present microtubes. Further development of the microcirculation system and various small vessel phantoms, including for deep brain microvessels, will be helpful for the diagnosis and future mechanistic study of lacunar infarction using 3T MRI.

## Figures and Tables

**Figure 1 sensors-22-06191-f001:**
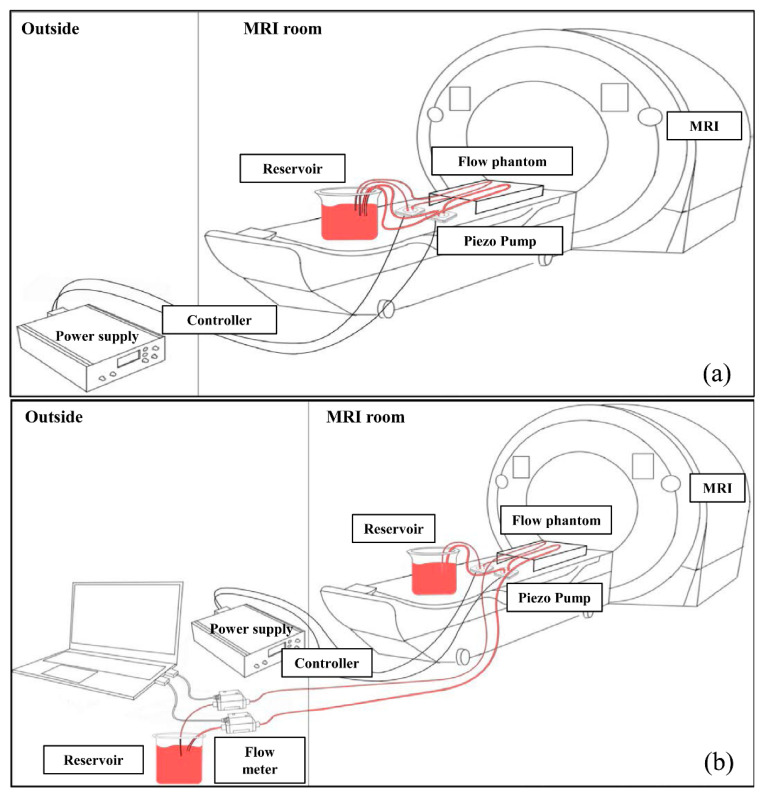
Flow system configuration using a 1 m tube (**a**) and a 6 m tube (**b**). (**a**) To confirm the structural shape of the microvascular phantom, the flow system was constructed by shortening the tube length to 1 m. (**b**) For flow velocities measurement, a flow system was connected by a 6 m tube to the flowmeter outside the MRI. Abbreviation: MRI, magnetic resonance imaging.

**Figure 2 sensors-22-06191-f002:**
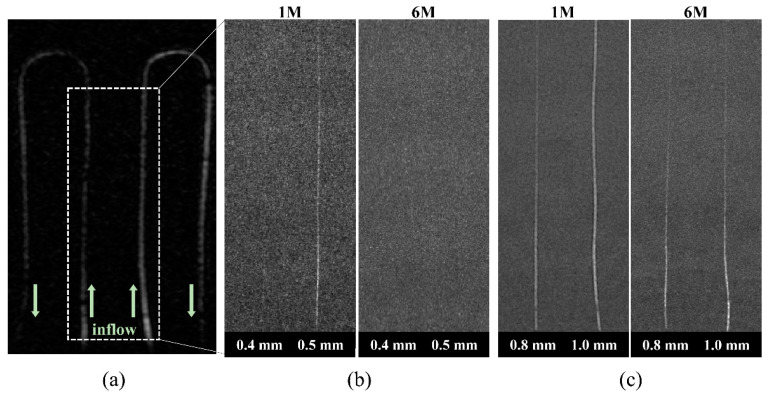
Three-dimensional high-resolution time-of-flight coronal maximum intensity projection (MIP) images for comparing diameters and lengths. (**a**) Coronal MIP image for reference. (**b**) MIP images of 1 m and 6 m tubes with 0.4 mm and 0.5 mm diameters. (**c**) MIP images of 1 m and 6 m tubes with 0.8 mm and 1.0 mm diameters.

**Figure 3 sensors-22-06191-f003:**
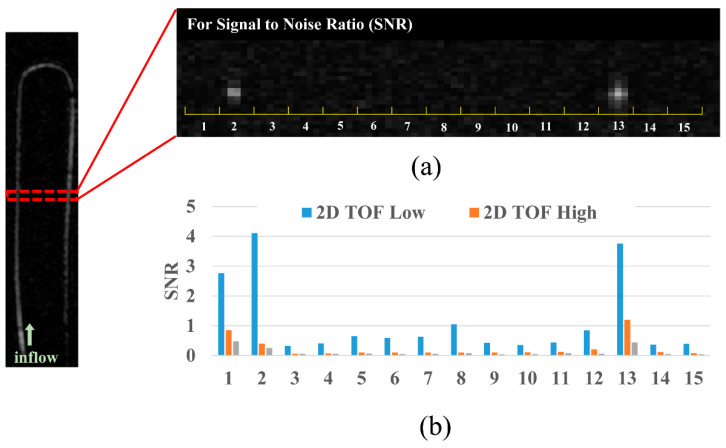
Comparison of signal-to-noise ratios (SNR) according to the magnetic resonance angiography sequences. A coronal maximum intensity projection (MIP) image in the 1.0 mm tube diameter is provided for reference (left). (**a**) Axial MIP image of the selected regions of interest. (**b**) SNR graph of each sequence. Note that these data are representative of 1.0 mm tube diameter. Abbreviations: TOF, time-of-flight.

**Figure 4 sensors-22-06191-f004:**
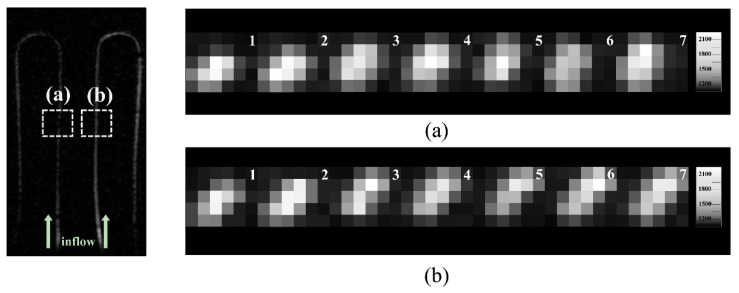
Microvascular phantom images for the reproducibility of the flow system. A coronal maximum intensity projection (MIP) image in the 0.8 and 1.0 mm tube diameters is provided for reference (left). (**a**) Seven axial MIP images of the 0.8 mm tube and (**b**) 1.0 mm tube.

**Figure 5 sensors-22-06191-f005:**
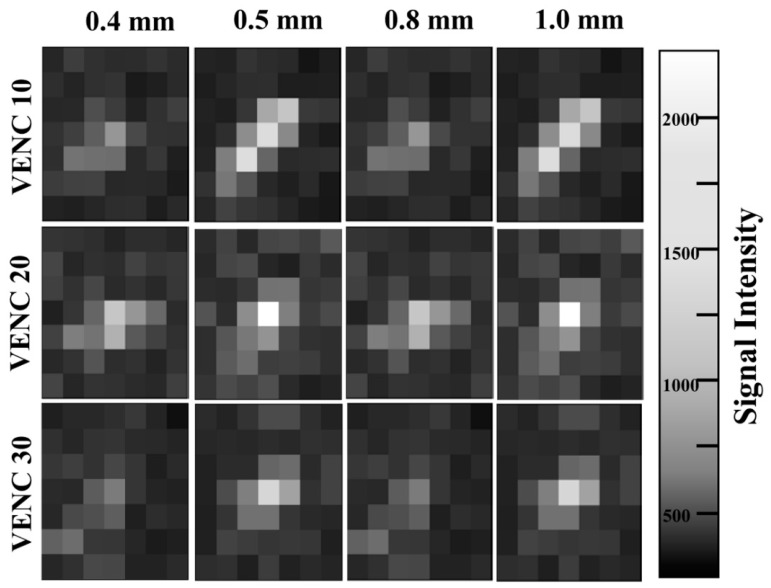
Comparison of flow velocities measured by 2D PC MRA with different velocity encodings (VENCs).

**Table 1 sensors-22-06191-t001:** Specifications of circulation systems used for vascular phantom studies.

	Targets	Diameter	MRI	Pump (Metallic)	Velocity or Flow Rate
Walsh (2005) [6]	Coronary artery	9 mm	1.5T	Pulsatile flow (Yes)	3.7 mL/s
Kim (2022) [10]	Circle of Willis	4.9 mm	3T	Peristaltic (Yes)	N-M
Ooij (2012) [11]	Anterior communicating artery	2.1 mm	3T	Centrifugal (Yes)	10.46 cm/s
Brindise (2019) [12]	Basilar artery	5 mm	3T	Gear (Yes)	3.09 mL/s
Azuma (2010) [13]	Internal carotid artery	5 mm	1.5T	N-M	20 cm/s
Wang (2020) [14]	Small artery	0.3 mm	7T	No pump (No)	N-M

N-M: Not mentioned.

**Table 2 sensors-22-06191-t002:** MRA sequence parameters.

Parameter	TOF				PC
	2D Low	2D High	3D High		
TR (ms)	20.0	20.0	30.0		28.6
TE (ms)	5.33	8.60	5.29		7.52
FA (deg)	30	30	25		70
Number of slices	60	30	44		10
Image plane	axial	axial	axial	coronal	axial
Matrix size	320 × 160	704 × 352	704 × 352	448 × 448	512 × 384
FOV (mm^2^)	176 × 88	176 × 88	176 × 88	175 × 175	174 × 130.56
TA (min:s)	1:28	1:27	2:26	3:01	1:00
BW (Hz/Px)	200	203	154	155	199
GRAPPA	Reduction factor = 2 with 24 autocalibration signal lines				
VENC (cm/s) Encoding direction	N/A				10/20/30 Through plane

Abbreviations: MRA, magnetic resonance angiography; TOF, time of flight; 2D, two-dimensional; 3D, three-dimensional; PC, phase contrast; TR, repetition time; TE, echo time; FA, flip angle; FOV, field of view; TA, acquisition time; BW, bendwidth; GRAPPA, generalized autocalibrating partial parallel acquisition; VENC, velocity encoding.

**Table 3 sensors-22-06191-t003:** Maximum signal intensities of tubes with various diameters and lengths in 3D High TOF sequence.

Tube Diameter (mm)	Maximum Signal Intensity	
	1 m	6 m
1.0	241	190
0.8	199	182
0.5	133	100
0.4	91	93

**Table 4 sensors-22-06191-t004:** SNRs of each TOF sequence at different tube diameters.

Tube Diameter (mm)	SNR		
	3D TOF High	2D TOF High	2D TOF Low
1.0	0.46	1.03	3.93
0.8	0.29	0.67	3.03
0.5	0.10	0.24	1.35
0.4	0.09	0.16	0.76

Abbreviations: SNR, signal-to-noise ratio; TOF, time-of-flight; 2D, two-dimensional; 3D, three-dimensional.

**Table 5 sensors-22-06191-t005:** Average signal intensity of seven PC 2D MRA images.

Tube ID (mm)	Signal Intensity							
	#1	#2	#3	#4	#5	#6	#7	Average ± SD
0.8	638	590	586	611	599	679	630	619.3 ± 32.7
1.0	636	588	648	658	604	605	627	624.0 ± 25.7

Abbreviations: PC, phase contrast; 2D, two-dimensional; MRA, magnetic resonance angiography; ID, inner diameter; SD, standard deviation.

**Table 6 sensors-22-06191-t006:** Flow velocities measured by 2D PC MRA with VENCs of 10 cm/s, 20 cm/s, and 30 cm/s and a flowmeter.

Tube Diameter (mm)	VENC 10	VENC 20	VENC 30
	Velocity of PC MRA (Flowmeter) (cm/s)		
0.4	1.13 (5.70)	2.47 (5.70)	2.56 (5.70) *
0.5	1.67 (5.69)	3.20 (5.43)	4.46 (5.52) *
0.8	3.72 (11.70)	8.50 (11.14)	10.03 (11.01) *
1.0	4.05 (8.55)	9.96 (8.32)	8.21 (9.04) *

Abbreviations: 2D, two-dimensional; PC, phase contrast; MRA, magnetic resonance angiography; VENC, velocity encoding. * indicates the minimal difference between the velocities of the PC MRA and flowmeter.

## Data Availability

The data presented in this study are available upon request from the corresponding author.

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
