# Peer review of "MRI-Compatible Microcirculation System Using Ultrasonic Pumps for Microvascular Imaging on 3T MRI"

_sensors, 2022, doi:10.3390/s22166191_

Round 1
Reviewer 1 Report
Comment No. 1: The Abstract should contain answers to the following questions: What problem was studied and why is it important? What methods were used? What are the important results? What conclusions can be drawn from the results? What is the novelty of the work and where does it go beyond previous efforts in the literature? Please include specific and quantitative results in your Abstract, while ensuring that it is suitable for a broad audience. References, figures, tables, equations and abbreviations should be avoided.
Comment No. 2: The originality of the paper needs to be stated clearly. It is of importance to have sufficient results to justify the novelty of a high-quality journal paper. The Introduction should make a compelling case for why the study is useful along with a clear statement of its novelty or originality by providing relevant information and providing answers to basic questions such as: What is already known in the open literature? What is missing (i.e., research gaps)? What needs to be done, why and how? Clear statements of the novelty of the work should also appear briefly in the Abstract and Conclusions sections.
Comment No. 3: An updated and complete literature review should be conducted and should appear as part of the Introduction, while bearing in mind the work's relevance to this Journal and taking into account the scope and readership of the journal. The results and findings should be compared to and discussed in the context of earlier work in the literature.
Comment No. 4: This paper should be edited grammatically.
Comment No. 5: Result and discussion section can be more improved from the physical point of view.
Reviewer 2 Report
The authors developed a microcirculation system that can be operated within a 3 T MRI. The method is interesting and the manuscript is well organized. I believe the experimental realization of this method is noteworthy. My comments and suggestions are given below:
1. In the experiment, the liquid is water, why not just use blood or something like blood.
2. If the tube has branches, what will be the result?
I think the authors may add some reply/explanations about the above two questions in the improved manuscript.
Overall, the manuscript provided a good method to obtain microvascular images, which can be accepted for publication after minor revisions according to the comments above.
Reviewer 3 Report
This work implements the compatibility of an ultrasonic pumped liquid flow system for microvascular imaging. The work done lacks novelty and does not meet the standards of the sensor journal to be published in the journal.
1- What is the originality of the article? It should be clearly stated in the introduction.
2- It is necessary to review the references provided and include some more in relation to the proposed work. Highlighting the novelty of the proposed work.
3- The results obtained should be compared with the bibliography introduced, highlighting the novelty and improvements of the proposed work.
4. Why is water used? Blood or with similar characteristics should be used to obtain a more realistic model of the images.
Round 2
Reviewer 1 Report
The authors have responded to my previous comments and revised the manuscript accordingly. I believe it is acceptable for publication in its form
Author Response
We appreciate your time and comments for reviewing the manuscript. Because of your valuable comments, the manuscript has been considerably improved through the revision process.

Reviewer 3 Report
I accept the authors' argument that this is preliminary work demonstrating the feasibility of using magnetic resonance imaging to visualize the microvasculature.
However, I have some additional comments on the paper:
1- the quality of the images in figures 1,2,3,3,4 and 5 should be improved.
2- What type of water was used in the study: salt water, mineral water or distilled water? justify in the paper.
Would there be any difference in the imaging system when using each of them?
3- A table should be included in section 4 comparing the advantages of the system proposed in the paper compared to other systems in the literature.
